# Towards Ending Immunization Inequity

**DOI:** 10.3390/vaccines9121378

**Published:** 2021-11-24

**Authors:** Anna Victoria Sangster, Jane M. Barratt

**Affiliations:** International Federation on Ageing, Toronto, ON M4K 2N1, Canada; jbarratt@ifa.ngo

**Keywords:** adult vaccination, immunization, equity, social determinants of health

## Abstract

Vaccine-preventable diseases (VPD) are responsible for a significant portion of mortality across the life course in both low-income countries and in medium- and high-income countries. Yet, countries are consistently below the adult influenza vaccination targets, with rates in recent times even falling in some areas. (1) The study Towards Ending Immunization Inequity seeks to understand the various factors that contribute to the accessibility and effectiveness of vaccine-related messages and campaigns including the effects of social determinants, with the knowledge that these opportunities for communication represent a unique policy lever to improving uptake rates of vaccination in the most at-risk communities. (2) To address this knowledge gap, a 3-phase mixed-methods study was conducted including a preliminary scan of existing vaccine schedules and NITAG recommendations, focus groups and a cross-sectional survey. (3) Study results indicated that social determinants play a key role in an individual’s knowledge of vaccine-related information including types of vaccines available, vaccination gateways, vaccine recommendations and vaccine safety. (4) However, knowing that social determinants can influence uptake rates does not readily create opportunities and entry points for governments to implement tangible actions. An accessible entry point to reducing and ending immunization inequity is through changes in public health messaging to reach those who are currently unreachable.

## 1. Introduction

Population ageing is unprecedented, as is the prevalence of people with non-communicable diseases (NCDs). Older people and those with chronic co-morbid conditions are at the greatest risk of being debilitated and dying from vaccine-preventable diseases such as influenza, pneumonia, and pertussis. The COVID-19 pandemic has laid bare the inadequacy of some health care systems to address those who are in most need of acute care, but moreover the critical importance of vaccines for all.

The COVID-19 pandemic has also brutally exposed the lack of preparedness of not only government but patient and ageing organizations to influence and activate policy change. The poorest of the poor now and in the future will pay the highest price if public health systems are not able to adequately respond to the unique barriers experienced at a local level. Equitable access to health care, including prevention interventions, reliable evidence-based information, and messages, is urgently needed to ensure no one is left behind.

### 1.1. Population Ageing and Non-Communicable Diseases

The rapidly growing prevalence and burden of NCDs are not only accelerated by population ageing, but as a population group, older people are by their very nature heterogeneous and further varied in terms of the accumulated impact of the physical and social environments on opportunities and health behavior. The relationship with our environment is also skewed by personal characteristics such as the family we were born into, our sex, our gender and our ethnicity, which can collectively lead to health inequities. The cumulative impact of these health inequities across the life course is often significant and underestimated in public health interventions such as vaccination.

By 2030, which marks the end of the UN Decade of Healthy Ageing (2021–2030), the number of people aged 60 years and older will be 34% higher than today, reaching 1.4 billion. By 2050, this population will have more than doubled to 2.1 billion and will outnumber adolescents and young people aged 15–24 years [1]. NCDs kill 41 million people each year, which is equivalent to 71% of all deaths globally [2]. In Europe, diabetes, cardiovascular diseases, cancer, chronic respiratory diseases, and mental disorders together represent ~86% of total deaths and 77% of the disease burden. In developed nations, approximately one in four adults have at least two chronic conditions, and more than half of older adults have three or more chronic conditions [3].

Rapid population ageing and the vulnerabilities that may prevail, together with the rise in chronic co-morbid conditions requiring complex health management, are driving an unprecedented demand for health and social care. Globally, lower respiratory tract infections (including influenza and pneumonia) resulted in more than 1.5 million deaths in adults 50 years or older in 2017 and accounted for 23 million years of life lost due to premature mortality [4]. Infectious diseases (such as influenza, pneumonia, shingles, diphtheria, tetanus, and hepatitis) increase the risk of hospitalization, disability and death among older people and are associated with a loss of functional ability and autonomy [5].

Maintaining and improving the health of ageing populations likely to have chronic co-morbid conditions are a game changer in managing the burden of care in acute and long-term care systems. The societal opportunities that arise from increased longevity largely depend upon maintaining and improving functional ability in older age. Additional years dominated by poor health, social isolation and dependency on care have negative implications for older people and the entire society.

Vaccine-preventable diseases cause long-term illness, hospitalization, and even death. In Europe, seasonal influenza causes between 4 and 50 million symptomatic cases each year and the death toll associated with influenza ranges from 15,000 to 70,000 every influenza season, in terms of excess deaths [6].

Vaccination is a frontline public health action and one of the most successful measures of modern times, preventing up to six million deaths worldwide every year—a staggering five lives each minute [7,8]. The World Health Organization (WHO) has rated it second only to clean water. The societal value of immunization is well known during childhood years. Unfortunately, and despite clear evidence in support of adult vaccination, access, availability, and therefore uptake rates are suboptimal in the adult population.

The targeted use of vaccines across the life course can contribute to generations of healthy older people yet there are unanswered questions about equity impacting the nature and effectiveness of campaigns and the intrinsic messages. Vaccination policy, in its broadest sense, must be crafted to reduce, rather than reinforce, these inequities.

### 1.2. Social Determinants and Immunization

Vaccine-preventable diseases (VPD) are responsible for a significant portion of mortality across the life course in both low-income countries and medium- and high-income countries [9]. Several vaccines such as those against influenza and pneumococcal pneumonia are specifically recommended for older people and those with underlying chronic conditions yet are neither fully utilized nor fully implemented by governments despite recommendations from National Immunization Technical Advisory Groups (NITAG) and associated scientific bodies [10].

The successful implementation of immunization programs depends on multiple factors. While biological, epidemiological, economic, and logistical factors related to vaccinations often invite the most attention, studies have shown that social determinants can also have a significant effect on immunization efforts [9].

The reasons for under- or non-vaccination of children from low- and middle-income countries convey important insights on the interconnected nature and impact of social determinants in routine childhood vaccination [11]. For example, a barrier to vaccination in low-income countries may be the physical distance to the clinic. This is also relevant for older people living in rural areas in high-income countries or older people isolated in urban communities. No public transportation system, inadequate schedules or the high cost of travel are systemic issues that result in individuals and communities being unable to access free and available health services including vaccinations [12].

Across the life course (from childhood to adulthood), education attainment is associated with health care utilization and health literacy [13]. People living in rural areas have fewer educational opportunities, which in turn is related to childhood immunization program utilization and extends to adulthood [14]. Additionally, childhood immunization rates are lower among certain immigrant groups, which may be due to language barriers, cultural and educational differences [15]. Research provides clear evidence that older migrants are also vulnerable to social isolation as they age in a foreign land and are at risk of greater health inequities [16].

Social determinants influence the accessibility of vaccinations and vaccination practices. In a systematic review of the health and influenza vaccination of people aged 65 years and older, it was found that the health system, the health care provider, and the patient were all determinants in the variations in coverage levels within countries [17]. Policies, practices, and vaccination strategies against seasonal influenza were found to be influenced by social determinants where vaccines are routinely available [17,18].

A universal set of messages will not by itself improve adult immunization rates and recent studies highlight the critical need for various tools and strategies commensurate with the characteristics of the populations [19]. Incorporating a framework that considers these determinants in vaccine policy design and implementation will foster immunization equity among the most at-risk populations against seasonal influenza and other respiratory vaccine-preventable diseases [20]. Governments can help to reduce inequity in adult vaccination through targeted messages conveyed in campaigns and policy decisions taken, including access to health care and prevention programs.

### 1.3. The Problem

Answering why countries are consistently below the adult influenza vaccination targets is a pressing issue as suboptimal rates persist within and among countries, with rates in recent times even falling in some areas. Some studies have identified modifiable barriers to influenza vaccination in older adults, such as the misconceptions of adult vaccination, the limited knowledge of existing immunization policies, and logistical issues related to vaccine delivery, including insufficient supply of age-specific vaccines, complex vaccination procedures, inability to determine timing and type of vaccination and lack of funding for vaccines or vaccine visits [17].

However, there is a more silent and insidious issue. In large part, campaigns to inform and encourage older people to have an influenza vaccination, and vaccination in general appear blind to the inequity caused through standard universal messages that do not consider social determinants [21].

Social determinants of health, such as income, education, literacy and access to and understanding of information, impact immunization uptake as well as general health outcomes across low–middle- as well as high-income countries [22]. No studies were found to have examined the impact of social determinants on messages and campaigns on influenza adult vaccination.

Findings from the International Federation on Ageing (IFA) study Changing the Conversation on Adult Influenza Vaccination reveals not only a paucity of targeted information but also a failure to address socioeconomic barriers to accessing, and understanding information needed to act on a decision to be vaccinated [21].

Influenza campaigns in France, Canada and Australia have similar messages to at-risk groups, which include pregnant women, persons aged 65 years and over to those with chronic conditions. Messages designed to respond to the unique characteristics of a subpopulation are not evident, thus making it difficult and a certain barrier to fully appreciate the serious nature of influenza. The UK and Germany have targeted messages differentiated for at-risk groups; however, the content and dissemination method are problematic [21]. Older people and those most vulnerable tend to be less likely to use the internet, which is problematic, with online information being the main communication channel used by public health authorities [23,24].

Barriers to vaccination among those most vulnerable to VPDs are generally not addressed through policies, governance, or program implementation. There is a clear social and economic case for reaching the unreached. Due to the disproportionate vulnerability and disease burden, vaccinating the unreached is the most cost-effective and has significant lifesaving and quality of life potential [25]. This is also true for new vaccines. Improving equity in adult immunization also opens the door to better coverage in other health care systems across generations. Immunization is often the intervention with the widest reach, and able to act as vehicle of delivery for other preventative measures.

Findings from the study Towards Ending Immunization Inequity seek to understand the various factors that contribute the accessibility and effectiveness of vaccine-related messages and campaigns including the effects of social determinants, with the knowledge that these opportunities for communication represent a unique policy lever to improving uptake rates of vaccination in the most at-risk communities. In doing so, it gives attention to the urgent need to fund recommendations from immunization bodies (e.g., NITAGs), increase vaccinator gateways (e.g., pharmacists and community nurses); and provide access to vaccines are tailored for specific populations.

## 2. Materials and Methods

It is well recognized that vaccine uptake is influenced by factors such as lack of recommendation by trusted sources including general practitioners, concerns about vaccine effectiveness and safety, a lack of knowledge and information, low perception of risk, difficult access to preventive activities, and socioeconomic predictors.

On the basis that rates of adult influenza vaccination have plateaued or decreased in recent times, it could be suggested that standard messages may play only a limited role in stimulating influenza vaccination rates among older people. No previous research examines the relationship between social determinants and campaigns and messages used to encourage older people and those most vulnerable to have their influenza vaccination.

### 2.1. Study Phases

#### 2.1.1. Preliminary Scan of Vaccine Schedules

To address this knowledge gap, a mixed-methods study was conducted with the first phase being desk research to gather data in each of the five selected countries (Canada, France, Germany, the UK and Australia) of NITAG recommendations which have/have not been implemented, vaccinator gateways (e.g., pharmacist, public health nurses, and mobile clinics) and vaccine schedules tailored for specific populations (e.g., older people as they are defined in the country and vulnerable populations). Information was gathered from publicly available government immunization schedules and supplemented as necessary by independent experts in instances where recent changes to immunization schedules were made but not yet reflected in government/public health information portals. Additionally, experts were consulted to address any gaps in publicly available information, most significantly information as it related to the cost of vaccines recommended by NITAGs but not publicly funded.

#### 2.1.2. Focus Groups

The second phase was to conduct focus groups optimally in each country to be studied using the findings of the desk research in the form of questions to gather information on ‘what influenza inequity looks like for older people, and those other populations at risk of VPD’ How respondents understand the topic, interpret questions and how to frame the topic/question in different ways could be particularly helpful in gathering information before developing a survey questionnaire.

The focus groups also represented an invaluable opportunity to understand how the concept of equity was conceptualized and operationalized at a country level allowing for a better understanding of how study findings could be leveraged, and subsequent actions executed in line with the capacity and practices of civil society.

A focus group guide was developed and shared with participants who attended a 60 min virtual focus group session via zoom teleconferencing software. These sessions included between 3 and 5 individuals to ensure that participants were able to engage and provide nuanced information given the complexity and breadth of questions posed.

#### 2.1.3. Cross-Sectional Survey

The third phase of this study was a cross-sectional survey to identify factors associated with vaccine hesitancy among older people, and the health care system more broadly among a sample of vulnerable populations in five countries. The target population of respondents was identified through a review of phase one results and aligned with associated NITAG recommendations in each of the 5 study countries. With the exception of the United Kingdom, where NITAG recommendations were for 50 years and over, respondents were aged 65 years and over.

The selection of survey sites (Canada, the UK, Australia, France and Germany) considered findings of the recent environmental scan, status of population ageing, availability and accessibility of national and provincial information and the existence of a comprehensive influenza immunization program. Findings from the previously conducted study entitled Changing the Conversation on Adult Influenza Vaccination (CCAV) illustrated that despite robust immunization programs, established NITAGs and extensive vaccine-related campaigns, Canada, the UK, Australia, France and Germany continued to report suboptimal adult vaccination rates [21]. The selection of these 5 high-income countries will allow for more targeted insight into underlying factors that contribute to the continued persistence of suboptimal vaccination rates while controlling for the potential confounding factors that would be likely in other countries due to the challenges associated with weak vaccination policy and associated national communication strategies that were observed in the aforementioned CCAV study. As appropriated in some countries (e.g., Canada), consideration was given to disaggregating the population by provinces or states and rural vs. urban areas.

Influenza vaccinations in all countries are recommended by governmental and advisory bodies for at-risk populations and funded under the National Immunization Program (NIP) or state/local immunization programs. Vaccination policies and campaigns are developed and updated on a regular basis suggesting that new evidence and report findings are more likely to be used and implemented in these settings.

A semi-structured cross-sectional survey was used to consider the associations between social determinants as they relate to influenza vaccination and vaccination more broadly. The design allowed for access to multiple exposures and outcomes and was co-designed and informed by various experts. A 20 min electronic survey with ~60 questions was developed and completed by 1000 participants per site online in each of the five sites.

The electronic survey of ~60 questions comprised both open-ended and closed-ended questions and was presented in lay language. Questions were reviewed and pilot tested before being finalized. Electronic surveys administered in France and Germany were fully translated into French and German, respectively, in order to maximize participation and mitigate the need for fluency/proficiency in English. Data were analyzed and a datafile and cross-tabulations were produced and then synthesized into clear reports from each of the 5 sites.

#### 2.1.4. Ethical Considerations

Ethical issues were taken into consideration to ensure no harm was done. Participation was voluntary, and respondents were given the opportunity to withdraw at any time without penalty. All data were de-identified to ensure confidentiality.

As the topic of interest relates to understanding the influence of social determinants on health inequity, barriers to participation were taken into consideration and steps made to mitigate these to the extent possible.

While the digital literacy of respondents was high across each of the study sites, these findings may not be representative of the digital literacy of all older adults more broadly. Additionally, the use of an online survey has implications within the context of equity.

## 3. Results

### 3.1. Demographics

Most respondents in Canada (60%) and Australia (56%) were in the 65–74 age groups, with the remaining being 75 years and older. In the UK, where eligibility for influenza vaccination is 50 years, just over half of those surveyed were 50–64 years, 39% in the 65–74 age group and 11% over 75 years of age. In Germany, nearly three-quarters (71%) were aged 75 years and older, whereas most (83%) were in the younger age group (65–74 years) in France. Slightly more participants were female (55%) and 45% were male. Most were heterosexual, married and/or common law, had internet access and were born in the country of study. While participants were generally fluent in the national language of the country in which they lived, between 10 and 30% were not able to speak in their native language to health care professionals (HCPs).

At least 50% of those in Canada, Australia and Germany had a chronic health condition, while the percentage was marginally less in France (46%) and the UK (39%).

Participants mainly lived in urban and suburban environments, with 20% in Canada and 40% in France living in a rural setting. The levels of education, while being varied, indicated that at least 1 in 3 people across all countries achieved a high school education or less. Most were retired from paid employment and one-third were in the lowest-income bracket.

With the exception of those in Germany, where approximately 20% of respondents indicated receiving either full- or part-time assistance, respondents generally lived independently in their own home and, if required, received low levels of assistance and services. Between one-third and one-half used an assistive device to improve or maintain their independence/autonomy.

When attending health-related appointments which were reported as largely unaccompanied, the primary mode of transportation was driving in Canada, Australia and the UK, whereas walking was the preferred mode in France and Germany.

More than 80% of those in Canada, Australia and the UK were comfortable asking questions of pharmacists, nurses and the general practitioner (GP), yet this was not always the case for those in France and Germany. Those in Germany were less comfortable asking questions of pharmacists (44%), nurses (52%) and GPs (35%), whereas the percentage was slightly lower (pharmacists 26%, nurses 34% and GPs 23%) in France. For the complete demographic profile of respondents, please refer to Appendix A.

### 3.2. Vaccine Awareness and General Knowledge

Self-reported general knowledgeable about vaccines and those recommended for older adults was generally high yet at least 15% had very little knowledge, were unsure of the safety, as well as the importance and necessity (especially at their age) for vaccines. In Canada, Australia and the UK, 15–20% reported not having enough knowledge to make an informed decision about their own health.

In France, less than half (47%) considered themselves knowledgeable about vaccines and slightly more (55%) knew something about vaccines for older populations. Between 45 and 53% did not have a view about vaccines and did not feel informed enough to make decisions about being vaccinated. This was particularly notable for participants who did not have a chronic condition, those who did not have a GP, were unaware of information about vaccines for older people, and most importantly those who had not been vaccinated since turning 65 years of age.

More than 3 in every ten people surveyed in France believe that vaccines are not important, nor safe and will not be beneficial to their health and well-being. Fifteen percent reported that the vaccine could potentially give them the disease it was created to prevent, while a further 29% were unsure of this statement.

Germany respondents have similar views about vaccine knowledge. One in four (25%) do not feel capable of deciding about being vaccinated based on their knowledge, and another 27% had no idea what to think.

As an emerging theme across all countries participants with a limited knowledgeable of vaccines and vaccination are often those who have not been vaccinated since turning 60 years of age, do not believe it is important for people in older age groups to be vaccinated, and/or who have not sought information about vaccinations for those in this age group.

Vaccine safety, efficacy and its value as an action to maintain and improve health and well-being is questioned by a not insignificant percentage of respondents when considering the population-based impact on health care systems. For example, 15% of Canadian respondents and 10% of those from the United Kingdom representing more than 1 million and 2.5 million citizens, respectively, do not believe that vaccines are safe, will not keep them healthy, and may even give them the disease they were created to prevent.

This situation is even more pressing in France and Germany. Fifteen percent of respondents (>1.7 million older adults) in France believe that a vaccine could potentially give them the disease and nearly one in three were unsure of whether the vaccine could give them the disease. In Germany, while most (60%) believe vaccines are both important and necessary for adults—and particularly for adults 60 years and over—19% believe that at this point in their lives, they have no use for vaccines, and two in ten believe vaccines are not safe and/or will not keep them healthy. One in four German respondents (>5.9 million) believe that vaccines might actually give them the infectious disease, with an additional 39% indicating that they were unsure whether this statement was true or not.

### 3.3. Knowledge of Vaccines Recommended for Older Adults

Vaccination throughout life is part of a comprehensive public health approach that considers the consequences of infectious diseases at various stages of life from birth to death. Awareness and accurate health-related information guides decision making and in the case of being vaccinated or not could be lifesaving.

Most respondents from Canada (92%), Australia (79%) and the United Kingdom (71%) identified at least one of three routine vaccines, namely, influenza, pneumonia and/or shingles without a prompt. This was not the case in France and Germany, where 49% and 32% identified only one of the three aforementioned vaccines.

Influenza was identified at least 28% more often compared with the other VPDs across the countries studied. This was true even in France and Germany, although 28% of German respondents identified influenza compared with just 13% and 6% for pneumonia and shingles, respectively.

Referring to a list of vaccines, the percentage of respondents identifying pneumonia and shingles vaccines as recommended for older adults increased significantly. For instance, there was a 25% increase in awareness of the pneumonia and shingles vaccine for respondents in the UK. In contrast, only 1% of those surveyed in France identified pneumonia and shingles vaccines as being recommended for older adults and this only moderately increased to 12 and 5% with additionally provided information.

A somewhat surprising finding was that 45 and 41% of those surveyed in France and Germany, respectively, identified the tetanus vaccine as recommended specifically for older adults compared with only 17 and 15% of those in the Canada and the UK respondents.

In the case of Germany, the higher rate of identification for the tetanus vaccine could stem from the increased incidence of tetanus that was seen as a result of the destabilization of the Russian medical system following the collapse of the Soviet Union in the period 1990–1997. As general hygiene conditions worsened during this time, cases of both diphtheria and tetanus rose, increasing the likelihood that cases could be acquired outside the country and then imported across its borders [26]. Furthermore, while cases of tetanus in Germany are rare when cases have arisen in recent years, they have been observed predominantly in unvaccinated older adults [27]. While the reason for the increased identification of tetanus in France is less clear, it is important to note that French health authorities explicitly recommend that anyone 65 years and older receive a booster for tetanus, whereas tetanus boosters are simply recommended at 10 year intervals in countries such as Canada and the UK [28,29,30].

### 3.4. Awareness of Vaccine Messaging

Targeted evidence-based messages should be at the core of all public health immunization campaigns to educate and inform the general public and those most at risk of vaccine-preventable diseases. Therefore, of concern is the finding that in Australia, 51% of respondents indicated that they had never seen, read, or heard anything about vaccinations specifically for older adults followed by Germany (41%), the UK (32%), France (28%) and Canada (27%).

Emerging trends across the countries highlighted factors, such as language and ethnicity, that may contribute to the awareness of vaccine-related messages. For example, in Canada and Australia, those whose first language was not English were less likely to have seen, read or heard any information about vaccinations specifically for older adults. In Australia, for example, 50% of those whose first language is English reported having been exposed to vaccine-related information compared with just 29% whose first language was not English. In Canada and the UK, language and ethnicity were also shown to reduce the comfort level of respondents in asking health-related questions and negatively impacted their ability to communicate effectively with health care professionals.

Findings also showed an association between vaccination history and vaccine awareness. In the UK, Australia and Germany, those who had not been vaccinated against any infectious disease since reaching the age of 65 years (50 years of age in the UK) were less aware of messaging compared with those who had been vaccinated. Age is also related to level of awareness, with younger respondents (those under the age of 65 years) in Germany, France, Australia and less than 50 years of age in the UK reporting lower levels of awareness compared with those over the age of 65 years.

Access (or lack thereof) to a GP was a further factor impacting level of awareness, alongside income and education. In Germany (24%) and France (27%), participants with a GP were more likely to have seen, read or heard vaccine-related information than those without a GP.

Although income and education levels were not associated with the awareness of vaccine messages across all countries, in France, those with an income of less than 60,000 Euro annually had lower levels of awareness than those with higher incomes. In Australia, those who had a university education or higher were 15% more likely to have been exposed to vaccine-related messaging compared with those with a college education or less.

### 3.5. Sources of Information

Understanding where and from whom individuals prefer to receive health-related information is crucial to improving uptake rates. Surprisingly, there was a reliance on media and advertising for information, which consistently ranked above influential people including doctors, nurses, pharmacists and specialists. Those in Canada and Australia used these channels to access information related to vaccination approximately 30% more than health care professionals.

Most believed that the general vaccination information available was relevant to their age and needs, yet far fewer felt confident about material that addressed questions about vaccines and their medical condition. For example, in the UK and Australia, only 4 in 10 surveyed believed that the vaccine-related information was specific to their medical condition and 50% from France and Germany believed this to be true.

### 3.6. GP Status and Comfort Asking Health-Related Questions

Non-medical factors, often referred to as social determinants, can significantly influence health outcomes. They are the conditions in which people are born, grow, work, live, and age, and the wider set of forces and systems shaping the conditions of daily life. These forces and systems also include economic policies and systems, development agendas, social norms and policies and political systems.

Access to affordable health services of decent quality, and health-related information can influence health equity. Overall, GPs were reported as the most influential source for information and health advice yet the ease with which individuals engage with these trusted health professionals is distinctly more difficult for those in two European countries. Between 90 and 98% of respondents in Canada, the UK and Australia had access to their GPs compared with only 80 and 66%, respectively, in France and Germany.

Typically, the entry point into the health care system is the family doctor, yet the study findings showed that certain social determinants may impact a person’s decision to be vaccinated. For example, in Canada and the UK language, and ethnicity were shown to reduce the comfort level in asking health-related questions and negatively impacted their ability to communicate effectively with health care professionals.

Confidence when making health-related decisions, and confidence that health care providers are looking after individual needs are important indicators of the strength and trust placed in health care professionals. In the United Kingdom, despite high levels of reported access to a GP (98%), one in five surveyed (20%) were not confident in making health-related decisions or relying on health care professionals. Similarly, 25% of respondents in France were unsure as to whether their HCPs were looking after their needs, and 13% reported they were not. In contrast, 58% of those in Germany felt that their needs were addressed by HCPs, with 63% confident and informed when making health-related decisions. Germans living in urban settings, those lacking a GP, and those living without a chronic condition were less confident.

Across all countries, those who had less confidence in health care professionals were more likely to not have been vaccinated since turning 65 years of age (and 50 years in the UK); had not been exposed to information about vaccines for older adults; and disagree with the importance of older adults getting vaccinated.

### 3.7. Vaccination Uptake and Gateway Information

Immunization across the life course is an effective public health intervention that benefits people of all ages. Until relatively recently, there has been an appropriate focus on pediatric immunization; sometimes it appears at the expense of the lives of older people who continue to contribute to the social and economic livelihood of their nations. Immunosenescence and weakened immunity undoubtedly place older people at risk of serious complications and even death from respiratory infections such as influenza, pneumococcal pneumonia, and pertussis and then shingles.

Part of understanding how to effectively address suboptimal adult vaccination rates is to acknowledge the baseline evidence and differences in current uptake rates that vary greatly from country to country. Eighty percent of respondents from Canada and 83% from Australia reported being vaccinated for influenza, pneumonia and/or shingles since turning the age of 65 years. In the European countries studied, the rates of vaccination of respondents are much less encouraging—only 70% of older adults in the UK, 62% in France and 57% in Germany have received any vaccination since turning the age of 65 (50 in the UK).

Motivation represents a key factor in improving uptake rates. Across all study countries, encouragement from others, especially a GP, prompted a person to be vaccinated. Additionally, there were striking similarities among those who had not been vaccinated. They were likely to have been not vaccinated against any infectious disease since turning the age of 65 years (50 years in the UK); did not have access to a family doctor; were younger than 70 years of age; free from chronic conditions; and either did not believe in the importance or were unaware of vaccines specifically recommended for older adults. In Australia, for example, respondents who did not have a GP were almost six times more likely to be unvaccinated (70 vs. 12%) and this trend was also evident in the UK where those without a GP were three times more likely to be unvaccinated.

Most of those surveyed in Canada, the UK and Australia indicated that the GP surgery is the preferred place to receive vaccinations, while only 62% in Germany and 40% in France indicated this location. Access to a GP, however, was strongly associated with knowledge of where to go to be vaccinated. The preference to be vaccinated at the surgery appears dependent on whether the respondent had access to a GP on a regular and routine basis. In Germany, for example, respondents who had a GP were 19% more likely to select the surgery as their preferred location to be vaccinated.

Interestingly, in Canada, respondents were just as likely to be vaccinated at a pharmacy and/or walk-in clinic to a GP surgery, whereas the preference appeared to be dependent on the region in the UK and Australia. For example, respondents from Wales, Scotland, or Northern Ireland were twice as likely to identify pharmacies or walk-in clinics compared with those from England. In Australia, respondents from Victoria and South Australia were more likely to rely on walk-in clinics than in other states and territories.

In France and Germany, a significant percentage (39 and 31%, respectively) indicated that they were unsure of where to go to be vaccinated. Factors leading to this response appear similar to those that impacted vaccine uptake rates. Individuals who had not been vaccinated for any infectious disease since turning the age of 65, did not have access to a family doctor, were younger than 70 years of age, free from chronic conditions and lacking general awareness of vaccines recommended for older adults were less confident about where they should go to receive important vaccinations.

## 4. Discussion

Vaccination represents one of the most effective public health interventions of our time and yet, despite the fact that countries around the world recognize this importance through national immunization schedules and vaccination targets, vaccination uptake rates remain shamefully low within and among countries. Understanding why this is the case is crucially important in identifying mechanisms through which vaccination rates could be improved. There is scant evidence of systemic evaluation of the impact of public health messaging and national vaccination campaigns specific to at-risk groups of older people and those with chronic conditions.

In 2020, the International Federation on Ageing (IFA) studied the content and nature of influenza campaigns in ten countries. Findings suggested that while each country utilized vaccination campaigns to encourage and increase uptake rates, the impact and efficacy of these campaigns was underwhelming with little or no impact on the uptake rates [21]. While it was recognized that there are many factors to improving coverage, the analysis revealed significant shortcomings related to the availability of targeted information while also highlighting a failure to address socioeconomic barriers to accessing, and understanding information needed to act on a decision to be vaccinated.

Social determinants of health can significantly impact immunization uptake as well as general health outcomes and yet no studies were found to have examined their impact on the accessibility or efficacy of vaccine-related messaging and campaigns among older adults. Disproportionate vulnerability and disease burdens which put certain individuals including older adults at increased risk of VPDs highlight the urgent need to prioritize vaccinating the unreached.

Improving equity in adult immunization not only has the ability to profoundly impact the health and quality of life of older adults but also represents a cost-effective solution and opportunities for better coverage in other health care systems across generations. Seeking to understand the effect of social determinants on the accessibility and efficacy of vaccine-related messages and campaigns among older adults represents a unique policy lever to improving uptake rates of vaccination in the most at-risk communities.

### 4.1. Statement of Principal Findings

To address this knowledge gap, IFA conducted a mixed-methods study which leveraged a cross-sectional survey to identify various factors that impact the accessibility and efficacy of vaccination messages and campaigns, and the health care system more broadly among a sample of vulnerable populations (older adults) in five countries (Canada, France, Germany, the UK and Australia). Study findings illustrated that despite generally high levels of general awareness, there is a significant subset of individuals across all countries that do not feel like they are knowledgeable enough about vaccines to make an informed decision. This is further exacerbated by the fact that awareness decreases precipitously when referring to vaccinations recommended specifically for older adults. Factors such as vaccination history, not having been vaccinated for any disease since turning 60 years of age, personal beliefs, not believing it is important for people in older age groups to be vaccinated, self-advocacy and not having sought out information about vaccinations for older adults all appear to play a significant role in general vaccine knowledge.

While vaccination history, personal beliefs and self-advocacy do not in and of themselves represent social determinants of health, they are undoubtedly impacted by broader social and structural determinants. There is strong evidence “that people who have more resources in terms of knowledge, money, power, prestige, and social connections are better able to avoid risk and to adopt the protective strategies that are available at a given time and a given place [31] p. 295.” Furthermore, personal beliefs about illness and treatment can be markedly different among ethnic minorities due to their cultural and religious beliefs, without culturally appropriate education this can contribute to the non-compliance with medical interventions that is seen more frequently among these populations [32].

Vaccine knowledge also appears narrow. Vaccines for influenza, a serious infection impacting older adults, were identified at least 30% more often compared with those for pneumonia and shingles were. This difference is likely due in part to the disproportionate prioritization of influenza by public health agencies [33]. While influenza, pneumonia and shingles are included in national immunization programs across all countries, governmental support and integration of NITAG recommendations are inconsistent. In Canada, for example, while the herpes zoster vaccine is recommended by the National Advisory Committee on Immunization (NACI), only one Canadian province (Ontario) provides public funding for this vaccine, highlighting the pressing need to prioritize not only the implementation of NITAG recommendations but also associated funding of these recommendations, without which vaccine uptake rates will likely remain suboptimal.

While the lack of association between income and vaccine awareness seen was somewhat surprising given the breadth of evidence regarding the relationship between income level and health literacy, it is possible that the reported incomes of study participants were not fully reflective of their previous earning potential [34]. With the exception of respondents from the UK, where the minimum age of participants was 50 years old, the majority of study participants reported their employment status as retired. As such, the reported incomes of respondents are likely less than their incomes while active members of the work force, and therefore potentially obfuscating the relationship between income and level of awareness.

### 4.2. Secondary Findings

The prioritization of influenza by national governments also has significant implications on the access and availability of vaccine-related messages. Given the fact that national campaigns are often developed by federal public health units, it is unsurprising that there is a disproportionate availability of information related to influenza compared to pneumonia and shingles, despite the significant risk all three VPDs pose to the health and well-being of older adults [21]. Furthermore, through CCAV study, it is known that universality and “sameness” of messages may form an impenetrable barrier for those with different levels of health literacy and at-risk populations [21]. This study confirmed this hypothesis, where respondents exposed to information specific to vaccines recommended for older adults (representing more tailored information) were more likely to view this as relevant to them and their medical conditions.

While large-scale campaigns are a powerful tool in promoting vaccination, study findings have also highlighted the crucial and influential role of HCPs and more specifically GPs in the provision of vaccine-related information. Access (or lack of access) to a GP alongside other important social determinants such as income, ethnicity, education and language have all been shown to play a significant role in a person’s ability to access vaccinations and understand related health information [20,35].

Despite high levels of access to a GP in Canada, the UK and Australia, there was still a significant subset who reported a lack of confidence when making health-related decisions, and confidence that health care providers were looking after their individual needs. Confidence dropped further in France and Germany, where the number of individuals who had access to a GP was considerably lower. This lack of confidence or mistrust of health care professionals is a complex and multifaceted issue; however, there is significant evidence that ethnic minorities or marginalized groups can often feel isolated due to language and cultural differences which leads to mistrust of health care professionals and poorer health care outcomes [36]. Language and ethnicity were also associated with reduced comfort level when asking health-related questions and negatively impacted ability to communicate effectively with health care professionals.

Additional findings have shown that individuals who do not have a GP had lower levels of awareness and knowledge related to vaccination and consequently significantly lower uptake rates compared with those with a GP. While there was limited information regarding the demographics of those who did not have a GP, there is reason to believe that marginalized groups experience disproportionate barriers when trying to access health care services [37]. This relationship could explain the differential access to GPs seen between Canada, Australia and the UK, where access was relatively high, compared with France and Germany, where GP access was markedly lower while the proportion of respondents who identified as an ethnic minority was significantly higher. This is particularly troubling as GPs were identified as both the number one source of information for vaccine-related information and the largest factor in motivating individuals to receive vaccinations, underscoring the urgent and crucial need to ensure policies and practices are put in place to reduce barriers and address the role of social determinants in the differential access to primary health care systems.

### 4.3. Study Limitations

Towards Ending Immunization Inequity represents the first study of its kind to have examined the impact of social determinants on the accessibility and/or efficacy of vaccine-related messaging and campaigns among older adults. While barriers to participation were taken into consideration and steps made to mitigate these to the extent possible, there are still inherent limitations associated with the utilization of online surveys.

In the context of social determinants, lack of access to digital technologies can represent a barrier to participation in this study, as could different levels of digital literacy. While reported levels of digital literacy among study participants were quite high, this could be somewhat skewed given that those with high levels of digital literacy may have been more inclined to participate additionally those with lower socioeconomic status may not have access to digital technologies that would allow for participation in this survey. Furthermore, while the survey was adapted and made available in the national language, individuals who may not be fluent in that language or have difficulty with literacy are likely to be underrepresented in this study. While this can be viewed as a limitation, study findings regarding access to and understanding of vaccine-related information appear highly dependent on fluency and therefore an assumption can be made that those with less fluency/literacy would likely face similar challenges in understanding vaccine-related messages and campaigns to those seen among study participants.

Another challenge is that the sample of study participants is not representative of the general population across study countries. In Canada, the UK and Australia, in particular, the proportion of Black, Indigenous, People of Color (BIPOC) individuals and those of different ethnicities was far lower than would be observed in the general population [38,39,40]. While this does have implications regarding the generalizability of study findings at a country level, the effect of this underrepresentation would likely only serve to further strengthen and highlight the existing findings around the deleterious impact of various social determinants on the access and efficacy of vaccine-related information.

Lastly, while the differing age of respondents was necessary to align with the associated NITAG recommendations in each country, respectively, the significantly lower minimum age of respondents from the UK (50 years of age versus 60–65) made direct comparisons between the UK and other study countries difficult. Given the finding that age was often a significant factor in vaccine awareness and behaviors, with younger individuals typically less engaged and knowledgeable about vaccine-related information than their older counterparts, the age of respondents in the UK likely contributed to the lower levels of vaccine awareness and uptake rates that were seen compared to similar countries such as Canada and Australia. For example, the lower rates of vaccination for influenza, pneumonia and/or shingles since turning the age of 65 years seen in the UK (70%) compared to Canada (80%) and Australia (83%) is likely due in part to the lower average age of respondent rather than the result of access to vaccine-related information or associated vaccination policy. In fact, while the reported uptake rates and levels of awareness in the UK are lower than those seen in Canada and Australia, the lower age range (reflective of NITAG recommendations) is indicative of more progressive vaccination policy for older adults and a greater access to publicly funded vaccinations for UK citizens.

### 4.4. Policy Implications

Evidence-based messages are at the core of public health campaigns developed and funded by government. These campaigns aim to educate and therein encourage citizens to be vaccinated against a range of communicable diseases. From the published results of the CCAV study, vaccination messages are universal, rather than shaped to speak to the specific needs of the most at-risk and most diverse populations. Current vaccine-related messages also fail to acknowledge the strong link between influenza and pneumococcal pneumonia. Given the strikingly low uptake rates and low levels of awareness seen for pneumococcal pneumonia and shingles vaccines, there is a clear need to better understand the factors that contribute to these rates (despite national recommendations for their use) as well as a concerted effort to increase and amplify existing vaccine messaging.

Governments appear to be expending substantial funds on public health campaigns which may not activate those most at-risk to be vaccinated. Greater attention to investment in campaigns in the broadest sense is likely to stimulate improvements in vaccination rates for the most at-risk populations (who are impacted already in the health system).

Findings from the Towards Ending Immunization Inequity study have highlighted the profound impact that underlying social determinants such as level of income, racial identity, culture and access to health services can have on an individual’s ability to receive and understand vaccine-related messaging. These findings represent a crucial foundation from which innovative campaign materials focused on capacity building and advocacy skills can be developed and gathered to support associations and agencies around the world prioritize adult vaccination.

## 5. Conclusions

The evidence is clear that social determinants play a critical role in health decisions and the ability for older adults to access and understand vaccine-related messaging and therefore receive vaccinations. However, knowing that social determinants influence uptake rates does not readily create opportunities and entry points for governments to implement tangible actions. Vaccination campaigns through public health messages are planned and executed with the primary goal of encouraging citizens to be protected against VPDs. An accessible entry point to reducing and ending immunization inequity is through changes in public health messaging to reach those who are currently unreachable. Existing funded campaigns serve as a critical platform for opportunities to change, and therein influence vaccination policy.

## Data Availability

The data presented in this study is contained within the Towards Ending Immunization Inequity article.

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
