# Peer review of "Towards Ending Immunization Inequity"

_vaccines, 2021, doi:10.3390/vaccines9121378_

Round 1

Reviewer 1 Report

I carefully read the paper and I found it very interesting and quite perfect as it is now. I really have compliments to the Authors.

Few minor points:

GP: please explain the first time it appears in the text.

Line 334. VPDs. It was already explaine din the text (line 87)

Line 385. Please clarify the sentence "Second to their GP across all countries 
between 60% and 80% of respondents favoured information from the television.". It sounds not clear in my opinion.

Reviewer 2 Report

The current study by Sangster et al. seeks to disclose the effect of social determinants on the accessibility and efficacy of vaccine related messages and campaigns among elder individuals. The authors conducted literature research on implementation of recommendations by National Immunization Technical Advisory Groups, vaccinator gateways and vaccine schedules for aged population groups in five high-income countries namely, Canada, United Kingdom, Australia, France and Germany, accompanied by a cross-sectional electronic survey on 1,000 participants per site. It has been claimed that social determinants, “such as individual and household income, education, literacy and access to and understanding of information” play a key role is an individual’s knowledge of vaccine related information however, this knowledge “does not readily create opportunities and entry points for governments to implement tangible actions”. The authors conclude the entry point to reducing the immunization inequity to be “through changes in the public health messaging” without specifying further.

First and the foremost, all the data obtained and referred to in the sections 3.2-3.7 of the manuscript, need to be summarized in table(s), similar to Demographics (3.1) in Appendix A. To make any specific conclusions evident for the reader, the trends need to be plotted as graphs/charts and discussed accordingly. For instance, each parameter (% or vaccinated people by specific vaccine type, % of immigrants, % of people with higher education, % people with income below and above $XX,XXX, % of people with official language proficiency, key characteristics of public health system etc.) vs. general vaccine awareness, belief in safety/unsafety of vaccines, confidence with information sources and level of trust to medical practitioners/television/media etc. Alternatively, the participants may be grouped according to the key parameters and the percentage calculated for each of the group.  The Discussion part needs to be rewritten to complete cover the data analysis and correlations found out. As of now, the manuscript looks a little too generalized and mire like a survey-based essay rather than the research article. Also, the original questionary should be provided in Supporting Information for review.

Second, the title is too vague and misleading and does not specifically reflect the manuscript topic. If the primary goal of the study was to identify “factors that impact the accessibility and efficacy of vaccination messages and campaigns, and the healthcare system more broadly” among designated population groups, the established links are neither evident not fully supported by the actual data except the access to general practitioners’ network. For instance, in Statement of Principal Findings the authors declare better recognition of influenza vaccines over the other ones likely, due to prioritized informational and funding support by federal and local agencies. How any of the social determinants other than simply prevalence of influenza over other vaccine preventable diseases may affect these decisions and policies making, has not been adequately explained.

Third, the reason behind the choice of the selected countries is not explicit – all of them of high-income country of comparable socioeconomic levels of development. A substantial portion of social barriers preventing from vaccination and characteristic for mid- to low-income countries is left behind. At the same time, while directly comparing UK/Australia/Canada to France/Germany it is likely, not taken into consideration that lower numbers may arise from the limited capacity of understanding the questions by non-native English speakers unless full translations were prepared.

Fourth, the introductive section is overall, poorly supported by referencing to the previous works particularly on Page 2, line 93, Page 3, line 119 and many others the statements made need to be supported by the appropriate citations. I am surprised, the authors ignored a recent comprehensive study by Roller-Wirnsberger et al., 2021 to mention about.

On Page 4, lines 154-156 the statement needs to be referenced too.

Fifth, when listing pneumonia along with the influenza infections, one should specify the bacterial and/or fungi origin of such (typically, pneumococcal but). For instance, “lower respiratory infections (including influenza and pneumonia)” suggests the authors may not fully understand the difference between pathological conditions and the causative agent(s).

Sixth, on Page 2 lines 76 and 78 when saying “Vaccination <…> preventing up to six million deaths worldwide every year” followed by “Immunizations save approximately 2.5 million deaths each year” sounds somewhat contradictive.

Seventh, it is unclear whether and how the comparison between UK and other countries takes into consideration the lower minimal age (50 vs. 65 y. o.) of population group selected for the study? This might have a notable impact on the rate of the responses given. 

Eighth, on Page 7, lines 344-347 the “surprising” observation of higher proportion of elder individuals in Germany and France to identify the tetanus vaccine as a recommended one might be explained by the historic influence of USSR vaccination policies through East Germany/GDR connections (Wiese-Posselt et al., 2011). On the contrary, on Page 8, lines 377-378 the findings on income and education levels not associated with the awareness of vaccine messages across all countries is quite surprising. Need to be discussed in more details further.

Minor comments and misspellings:

GP abbreviation needs to be deciphered when using first time in the manuscript text.

On Page 1, line 15 “social determinants play a key 14 role is in an individual’s knowledge

On Page 1, lines 39-40 what did the authors mean by “structural inequities” in this context?

On Page 1, line 40 “As a population group,“ comma needed

On Page 2, line 54 “represent ~86% per cent of total deaths” or “represent ~86% percent of total deaths

On Page 2, line 73 “the death toll associated with influenza is ranges from between 15,000 and 70,000” or “the death toll associated with influenza is ranges from 15,000 and 70,000” 

On Page, lines 89-90 “Several vaccines such as those against influenza and pneumococcal pneumonia specifically target older people” I am unsure whether flu or pneumonia vaccines “specifically target older people” (whatever it could be meant by the authors) but rather being recommended or prescribed additionally. 

On Page 2, line 91 “yet are not neither fully utilized nor fully implemented by governments

On Page 3, line 114 “Social determinants influence the availability and vaccination practices” availability of what?

On Page 3, line 115 “In a systematic review of the health and influenza vaccination of people aged 65 years of age

On Page 4, lines 159-160 “Barriers to vaccination among the most vulnerable are generally not addressed through policies, structures, governance, or program implementation.” Most vulnerable whom? What structures are authors talking about?

On Page 4, lines 167-169 the entire sentence sounds confusing. Consider revising

And overall, an extensive spellcheck and stylistics editing is requited for the manuscript.

Author Response

All comments have been responded to and can be seen in the attached document. One pending comment that the authors required further clarification regarding was the comment listed below.

when listing pneumonia along with the influenza infections, one should specify the bacterial and/or fungi origin of such (typically, pneumococcal but). For instance, “lower respiratory infections (including influenza and pneumonia)” suggests the authors may not fully understand the difference between pathological conditions and the causative agent(s).

The authors would be happy to provide the relevant clarification but cannot find the passage that the reviewer is referencing and no specific location has been provided. 
